# Influence of Posture and Coil Position on the Safety of a WPT System While Recharging a Compact EV

**Valerio De Santis** [1,*] **, Luca Giaccone** [2] **and Fabio Freschi** [2]

1 Department of Industrial and Information Engineering and Economics, University of L'Aquila, 67100 L'Aquila, Italy

2 Dipartimento Energia "G. Ferraris", Politecnico di Torino, Corso Duca degli Abruzzi, 24, 10129 Torino, Italy; luca.giaccone@polito.it (L.G.); fabio.freschi@polito.it (F.F.)

* Correspondence: valerio.desantis@univaq.it

**Abstract:** In this study, the human exposure to the magnetic field emitted by a wireless power transfer (WPT) system during the static recharging operations of a compact electric vehicle (EV) is evaluated. Specifically, the influence of the posture of realistic anatomical models, both in standing and lying positions, either inside or outside the EV, is considered. Aligned and misaligned coil configurations of the WPT system placed both in the rear and front position of the car floor are considered as well. Compliance with safety standards and guidelines has proven that reference levels are exceeded in the extreme case of a person lying on the floor with a hand close to the WPT coils, whereas the system is always compliant with the basic restrictions, at least for the considered scenarios.

**Keywords:** electric vehicle; EMF safety; numerical dosimetry; wireless charging; wireless power transfer





## 1. Introduction

Due to the increasing environmental concerns, renewable energy sources have recently attracted a great deal of attention from both industry and academia [1]. A key technology following this trend is the usage of electric vehicles (EVs), whose widespread diffusion is still limited by the charging infrastructure and their on-board energy storage systems, mainly batteries [2]. To overcome so-called "range anxiety", static or dynamic wireless power transfer (WPT) systems have been proposed to recharge EVs either while they are parked or in movement [3]. However, one of the main issues related to EV-WPT systems is the large electromagnetic field (EMF) emissions during recharging operations. Indeed, the demand for fast charging has increased the power level of WPT systems from 3.3 up to 22 kW [4], yielding an EMF leakage larger than in conventional wireless systems used to recharge consumer devices. This leakage in the neighborhood environment of the car (outside and inside) has increased the need to determine the compliance of WPT systems with international safety standards and guidelines [5,6].

The exposure assessment of static and dynamic EV-WPT systems has been widely investigated [7–15]. However, while the influence of the car chassis material has been investigated in [14,15], the effect of the human posture and related positions against the WPT coils has not rigorously been addressed. Such an influence is therefore investigated in this work for a large variation of anatomical models, postures and WPT coil position/configurations. Specifically, the magnetic field emitted by a static WPT system operating at the intermediate frequency (IF) of 85 kHz and engaged in recharging the battery of a compact car, namely a FIAT 500, has been considered.

The compliance assessment of EV-WPT systems is not straightforward. Indeed, while the standalone design of the recharging system could be easily performed with classical numerical approaches, the presence of the car body, which is more difficult to take into account, has been shown to play an important role [14,15]. However, the presence of

the human body does not affect the source field up to some megahertz [16,17], making it possible to separate the overall compliance procedure into two steps: (1) the simulation of the magnetic field source (WPT system and car body) and (2) numerical dosimetry (human body subject to the previously evaluated IF field). Step (1) is solved with an ad-hoc hybrid scheme coupling the boundary element method (BEM) with the surface impedance boundary conditions (SIBCs) in order to fit both the multiscale open-boundary (WPT system) and thin-sheet (car body) characteristics of the problem [18]. Step (2) is instead performed with the commercial software Sim4Life (https://zmt.swiss/sim4life, accessed on 26 October 2021), which relies on a Virtual Population (ViP). This allowed us to achieve the non-trivial task of assessing the numerical dosimetry on realistic anatomical models with different postures resembling those of a driver, of a person lying on the ground floor or in the rear-seats and of bystanders near to the car, while the WPT coils (both aligned and misaligned) were placed either in the rear or front position of the car floor due to the presence of the battery pack between the wheels.

## 2. Materials and Methods

### 2.1. Car Modeling

The compact vehicle considered in this paper is the FIAT 500, as described in [15] and freely accessible at this link https://github.com/cadema-PoliTO/vehicle4em (accessed on 26 October 2021). Once again, onlythe chassis of the car was considered and modeled as a surface mesh in order to exploit the capabilities of the numerical formulation, which is based on the hybrid BEM/SIBC method [18].

In contrast to [15], where the material properties of aluminum and carbon fiber were selected for the car body, in this study, a conductivity of $\sigma = 2 \times 10^6$ S/m and a relative magnetic permeability of $\mu_r = 300$ have been adopted. These values correspond to common steel with moderate shielding capabilities, as suggested in [19].

### 2.2. WPT System Configuration

In this paper, we have considered a WPT system classified as WPT2/Z3 by the standard SAE J2954 [4]. The input power is set to 7.7 kVA and the operational frequency is fixed to 85 kHz. For the assessment of these kinds of WPT systems, a time-harmonic formulation is sufficient because the harmonic content is negligible [19]. Furthermore, it is possible to assume a continuous sinusoidal wave even in the case that the actual waveform would be a sinusoidal burst [10].

The clearance between the receiving coil and the ground is set to 200 mm. Each coil is made of 8 turns, and the current flowing into a single turn is 26 A for the transmitter and j26 A for the receiver. Both coils are shielded by two thin layers of aluminum and ferrite with an outer dimension of approximately $420 \times 420$ mm$^2$, as shown in Figure 1.

In order to investigate the worst exposure scenario, both the case of perfect alignment (see Figure 1a,b) and of maximum misalignment, as suggested by SAE J2954 [4]—i.e., $d_x = -75$ mm and $d_y = 100$ mm (see Figure 1c,d)—were considered.

In contrast with [15], where the WPT system was placed below the car floor on the driver's side, two different locations were selected: one under the bonnet (see Figure 2) and the other under the baggage compartment (see Figure 3). This was done to avoid interference with the battery pack, which is normally placed between the rear and front wheels.

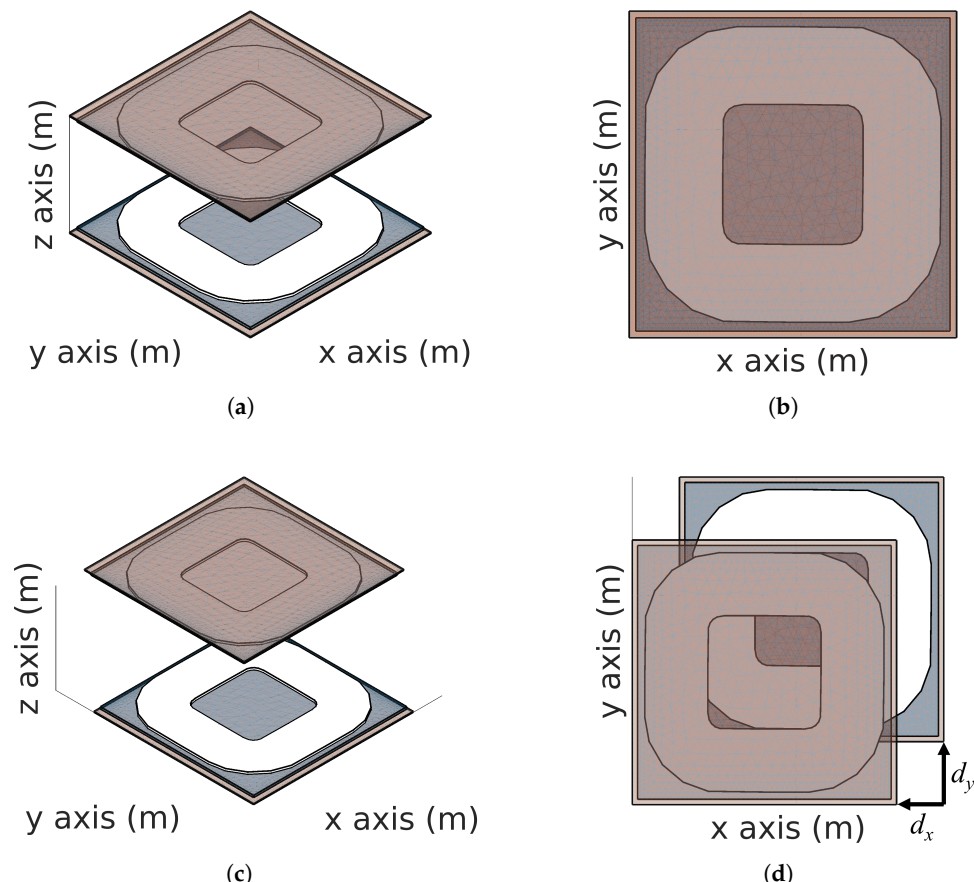

**Figure 1.** WPT2/Z3 system with perfect alignment (**a**,**b**) and maximum misalignment (**c**,**d**).

### 2.3. Exposure Scenarios for Numerical Dosimetry

Compared to [15], where a driver and a bystander were investigated (called exposure scenario #1 from here on), two further exposure scenarios, each consisting of two realistic anatomical models and two coil positions, have been considered, as illustrated in Figures 2 and 3, respectively. In particular, exposure scenario #2 consists of two adult males: one lying on the ground floor with a hand stretched towards the coils (worst-case scenario) and the other standing in front of the car. Exposure scenario #3 consists of two females: one child sleeping on the rear seats and one adult standing at the back of the car. All the anatomical human models are taken from the ViP 3.0 provided by the IT'IS Foundation (https://itis.swiss/virtual-population, accessed 26 October 2021), with a posable model for Duke lying on the floor.

Tissue dielectric properties of the human models were assigned from the IT'IS database [20], with the exception of the skin, where a higher conductivity value was adopted, as described in [21]. A uniform grid size of 2 mm was used to discretize the computational domain embedding the anatomical models (see Figures 2a and 3a).

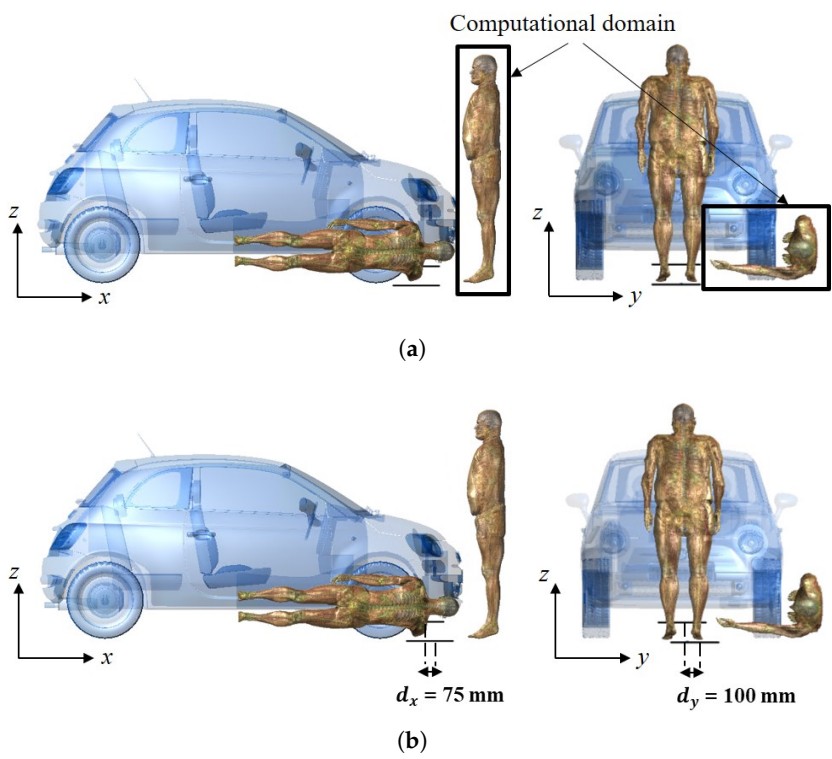

**Figure 2.** Exposure scenario #2: Duke lying on the ground floor and Fats standing in front of the car for the aligned (**a**) and misaligned (**b**) coils.

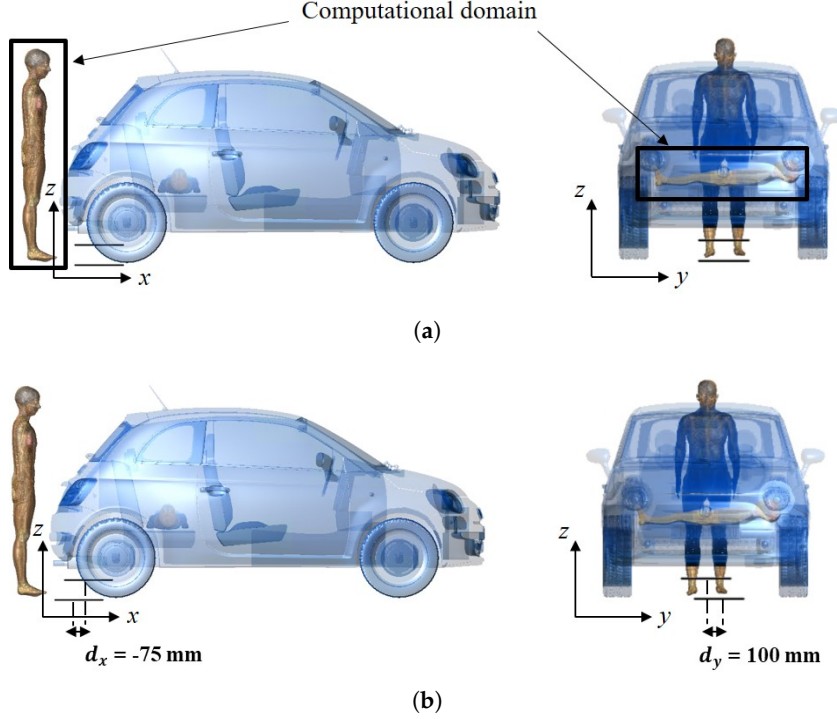

**Figure 3.** Exposure scenario #3: Roberta sleeping on the rear seats and Ella standing at the back of the car for the aligned (**a**) and misaligned (**b**) coils.

### 2.4. IF Dosimetry

It is well known that numerical dosimetry up to intermediate frequencies (10 MHz) can exploit the fact that the induced currents in the human body do not perturb the external

magnetic field [16,17]. For this reason, the simulations of the external magnetic field source can be separated by the evaluation of the electric field induced inside the human bodies. This division makes it possible to select the most suitable formulations for the two steps.

In this paper, the simulation of the WPT system and the car body is handled by a numerical hybrid formulation based on BEM and SIBC methods [18]. This formulation is particularly suitable to handle the multi-scale problem as the car body has a significant surface with a very small thickness.

The numerical dosimetry computations are instead performed using the Scalar Potential Finite Element (SPFE) method, which is implemented in the commercial software tool Sim4Life. Based on the magneto-quasi-static (M-QS) approximation and the conduction-current-dominant characteristics of biological tissues in the IF region, a simplified scalar potential equation is given by

$$\nabla \cdot \sigma \nabla \phi_e = -j\omega \nabla \cdot \sigma \mathbf{A} \tag{1}$$

where $\mathbf{A}$ is the magnetic vector potential, $\phi_e$ is the scalar electric potential, $\omega$ is the angular frequency, and $\sigma$ is the conductivity. Due to the fact that the magnetic field source is handled by a hybrid formulation based on BEM/SIBC, we cannot directly compute the necessary magnetic vector potential $\mathbf{A}$ on the right hand side of Equation (1). Therefore, the magnetic flux density $\mathbf{B}$ is computed via step (1), and a compatible magnetic vector potential $\mathbf{A}$ is then evaluated by using one of the curl-inversion procedures described in [22–24]. Specifically, Sim4Life implements the curl-inversion procedure based on Laakso et al. [22], though different schemes can be exploited by providing an external text file.

Once the magnetic vector potential $\mathbf{A}$ is provided, Equation (1) is discretized using the Galerkin Finite Element Method and linear nodal basis functions on a rectilinear grid. The resulting linear equation system is then solved using a conjugate gradient solver with a stopping criterion of 10 orders of magnitude reduction for the initial residual. Upon solving the unknown scalar potential $\phi_e$, the induced electric field $\mathbf{E}$ can be computed from

$$\mathbf{E} = -\nabla \phi_e - j\omega \mathbf{A}. \tag{2}$$

## 3. Numerical Dosimetry Results

The aforementioned two-step approach is hereby undertaken to conduct the compliance assessment of the investigated WPT system against the EMF limits for the general public provided by the International Commission on Non-Ionizing Radiation Protection (ICNIRP) [6]. First, the $\mathbf{B}$-field is computed outside and inside the car (without the human models) by means of step (1) and compared with the reference level (RL). Then, by means of step (2), the $\mathbf{E}$-field induced inside the human body is evaluated for comparison with the basic restriction (BR).

### 3.1. RL Numerical Dosimetry

Figures 4 and 5 illustrate the magnetic field distributions (both aligned and misaligned coil configurations) inside the computational domains of the considered exposure scenarios #2 and #3, respectively. In these figures, the anatomical models are overlaid on the exposure scenario only for the sake of clarity; i.e., to facilitate the understanding of the compliance. As can be observed, the ICNIRP-RL is never exceeded in the sleeping Roberta model. Instead, it is barely (aligned) or moderately (misaligned) exceeded in the feet of the standing models (both front and back of the car) and is greatly exceeded in the hand (up to the wrist area) of the lying Duke model. Thus, compliance with BR is necessary only in these latter cases where the RLs are exceeded.

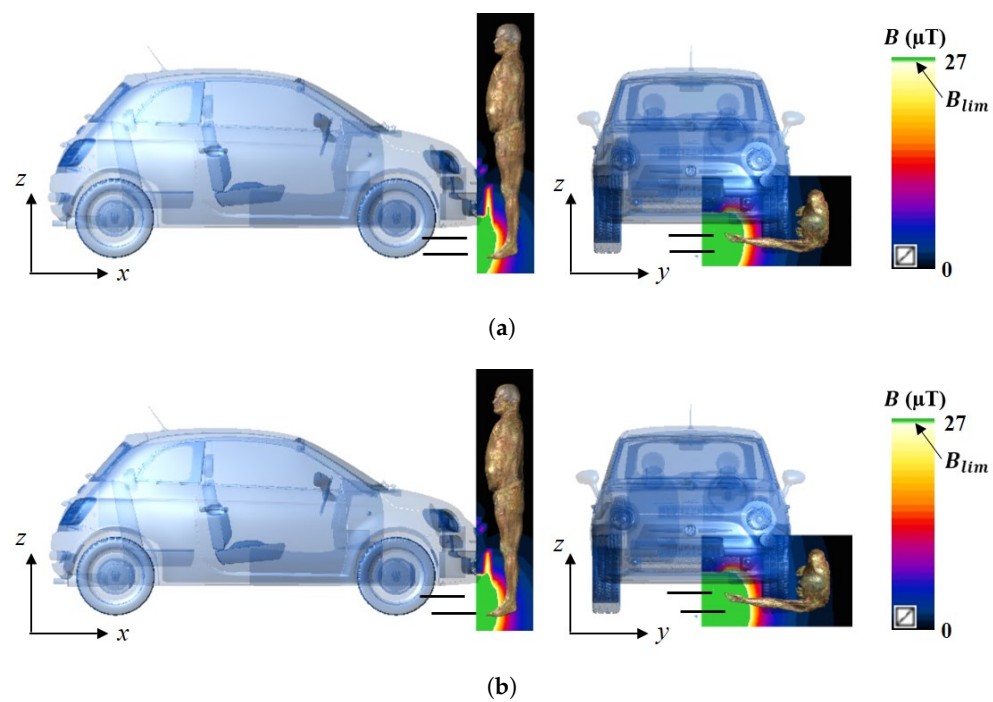

**Figure 4.** **B**-field distributions for exposure scenario #2 in the aligned (**a**) and misaligned (**b**) coil positions. $B_{lim}$ is the RL = 27 μT (the green area is the portion where the RL is exceeded).

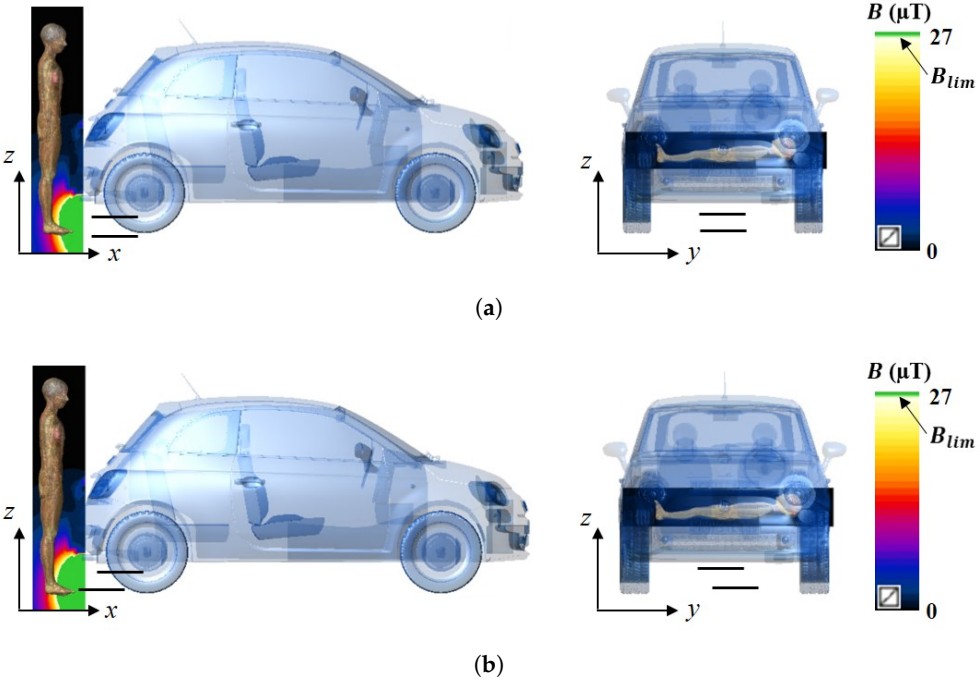

**Figure 5.** **B**-field distributions for exposure scenario #3 in the aligned (**a**) and misaligned (**b**) coil positions. $B_{lim}$ is the RL = 27 μT (the green area is the portion where the RL is exceeded).

### 3.2. BR Numerical Dosimetry

The induced electric field distributions inside the different anatomical models for both exposure scenarios (#2 and #3) and both coil positions (aligned and misaligned) are reported in Figures 6 and 7, respectively. These figures show that the ICNIRP-BR is never exceeded, except for a small portion of the wrist when the lying posture on the ground floor is considered. However, it is worth noting that ICNIRP suggests determining compliance against a $2 \times 2 \times 2$ mm$^3$ average volume and the 99th percentile of the peak induced

electric field [6]. In this work, anatomical models with a voxel resolution of 2 mm have been considered, and therefore the only 99th percentile has to be computed. Nevertheless, the 99.9th percentile is evaluated as well since the 99th percentile has sometimes been shown to underestimate the compliance, especially in the case of localized exposures [13,25–30].

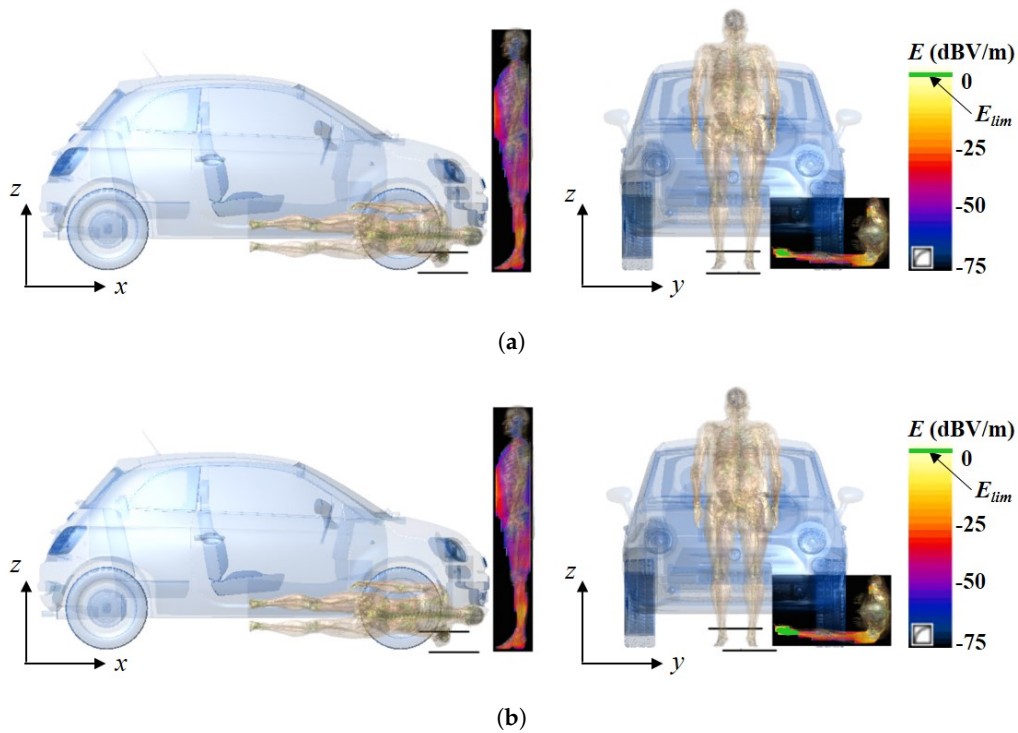

**Figure 6.** E-field distributions for exposure scenario #2 in the aligned (**a**) and misaligned (**b**) coil positions. $E_{lim}$ is the BR = 11.48 V/m (the green area is the portion where the BR is exceeded).

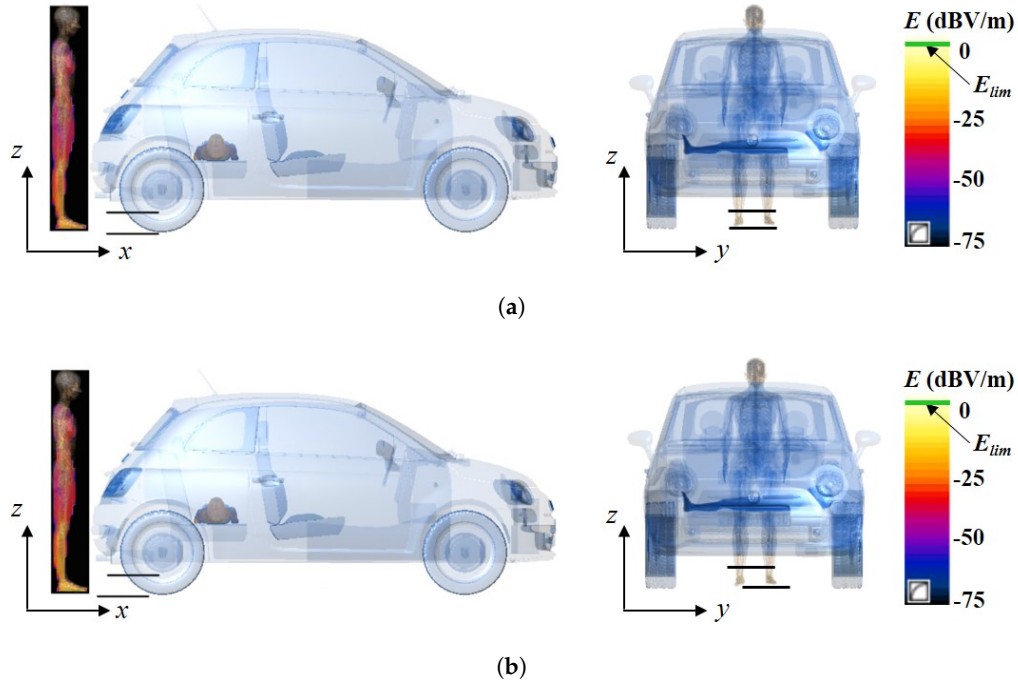

**Figure 7.** E-field distributions for exposure scenario #3 in the aligned (**a**) and misaligned (**b**) coil positions. $E_{lim}$ is the BR = 11.48 V/m (the green area is the portion where the BR is exceeded).

To better quantify these results, the values of the exposure assessment are summarized in Table 1, where $E_{max}$ is the peak induced electric field, whereas $E_{99.9}$ and $E_{99}$ are the 99.9th and 99th percentiles, respectively. As can be observed, when comparing the latter with the BR, the overexposure is always negative (at least $-6$ dB), meaning that the considered exposure scenarios are far from exceeding the ICNIRP-BR.

**Table 1.** Summary of the compliance with the BR for the considered exposure scenarios.

| Exposure Scenario | Chassis Material | Coil Position | $E_{\mathbf{max}}$ (V/m) | $E_{\mathbf{99.9}}$ (V/m) | $E_{\mathbf{99}}$ (V/m) | Overexposure (dB) |
|---|---|---|---|---|---|---|
| #1 (from [15]) | aluminum | Aligned | 8.26 | 1.84 | 0.71 | −24.17 |
| | | Misaligned | 7.69 | 1.36 | 0.57 | −26.08 |
| | carbon fiber | Aligned | 19.21 | 5.86 | 1.71 | −16.48 |
| | | Misaligned | 24.01 | 6.94 | 1.76 | −16.23 |
| #2 | steel | Aligned | 31.84 | 10.39 | 5.65 | −6.15 |
| | | Misaligned | 33.51 | 10.52 | 5.73 | −6.03 |
| #3 | | Aligned | 1.30 | 0.39 | 0.21 | −34.75 |
| | | Misaligned | 2.68 | 0.61 | 0.28 | −32.25 |

## 4. Conclusions and Discussions

In this paper, the influence of posture and coil position on the human safety of a WPT system engaged in recharging a compact electric vehicle was studied. The dosimetric analysis was performed by selecting a suitable mix of tools capable of analyzing the magnetic field source and evaluating the induced electric fields. The former was handled by ad-hoc software based on a hybrid scheme, whereas the latter was carried out using commercial software. This combination allowed us to handle the complex shape of the compact vehicle (namely a FIAT 500) and realistic anatomical models with different postures in a seamless way.

In order to investigate the effect of the posture and body–coil positions, a large variation of anatomical models (age, sex and body mass index) and exposure scenarios have been considered. Specifically, different postures resembling those of a driver, a lying person on the ground floor or rear-seats and bystanders near to the car were employed, while the WPT coils (both aligned and misaligned) were placed below the car floor before either the rear or front wheels due to the presence of the battery pack between the wheels.

From the analysis of the obtained results, it has been shown that the considered exposure scenarios are not compliant with the reference level, especially for a lying person with their hand close to the WPT system. Instead, compliance with the basic restriction is always satisfied, at least for the considered cases. In future, different exposure scenarios could be investigated, including heavier vehicles, such as SUVs and buses, or different anatomical models for the same exposure scenario. In the former cases, a higher power of the WPT system, together with a taller car floor, would lead to larger EMF leakages, whereas in the latter cases, different postures or anatomical details could yield higher induced fields.

Finally, it is worthy of mention that the influence of the chassis material could play a relevant role on the exposure assessment. While current steel with moderate shielding capabilities has been considered in this work, lower shielding performances have been found in previous papers by the authors when considering a futuristic chassis made of composite materials.

**Author Contributions:** The authors contributed equally to this work. All authors read and agreed to the published version of the manuscript.

**Funding:** This research received no external funding.

**Institutional Review Board Statement:** Not applicable.

**Informed Consent Statement:** Not applicable.

**Data Availability Statement:** Not applicable.

**Acknowledgments:** The authors would like to thank Donato Manesi (Politecnico di Torino) for the support given in the CAD modeling and Alessandro Franceschini (University of L'Aquila) for his valuable support with the dosimetric simulations.

**Conflicts of Interest:** The authors declare no conflict of interest.

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
