# Peer review of "Influence of Posture and Coil Position on the Safety of a WPT System While Recharging a Compact EV"

_energies, doi:10.3390/en14217248_

Round 1

Reviewer 1 Report

This is a well-conducted, well-described dosimetric study of personal magnetic field exposure for wireless power transfer to an electric vehicle, a highly relevant topic in view of the worldwide energy transition. I have only a few textual suggestions to clarify some (potentially complicating) issues that are not yet discussed.

Introduction:

What is meant by "conventional wireless systems"?

Materials and methods:

WPT system configuration: Is the 85 kW WPT field sinusoidal (or considered to be sinusoidal for simulation purposes)? If not, that is if has a  more complex waveform or is pulsed, the B-field strength can not be directly compared with the reference level and more complex assessment may be necessary (e.g. using ICNIRP's 2003 guidance on non-sinusoidal waveforms).

Results/Discussion:

A complicating factor in judging the exposure levels compared to the ICNIRP limits is that different human models were used for different exposure scenarios, i.e. two variables were changed at the same time. For the highest exposure scenario (Duke lying on the ground next to the car), it should be discussed how likely it would be that the basic restrictions are also met for the smaller human models (Ella and Roberta).

It would also be useful to add a brief discussion to what the degree the results could be generalized to other exposure scenarios, or what other scenarios it would be useful to investigate, for example larger cars or an electric bus.

Minor textual points:

Scenario #2, position described as "car floor" or "lying on the floor": should be changed to "lying on the ground next to the car", to prevent confusion with the situation where a person is lying on the floor of the cabin (inside the car).

p.2: "baggage car" should be "baggage compartment" ?

p.3: "two more different" should be "two different"; "one children" should be "one child"

p.4: "does not perturb" should be "do not perturb"

p.8: "less shielding performances" should be "lower shielding performances"

Reviewer 2 Report

The paper is interesting and helpful in practice,

The fluency of the text in the Introduction must be improved,

Please move the provided links in the text to the reference list,

Since the paper conducts a compliance test, please provide some detailed conclusions, maybe some bullet points, in the conclusion and any suggestions for improvements 

Reviewer 3 Report

The article is about the safety of the WPT system while recharging a compact car. This study is unique work to attach many people who are in this field. It could be nice if these tests are also repeated with SUVs or other cars. Great work! Thanks.
